Economic costs of invasive rodents worldwide: the tip of the iceberg

Diagne Christophe chrisdiagne89@hotmail.fr 1 2
Ballesteros-Mejia Liliana 2
Cuthbert Ross N. 3
Bodey Thomas W. 4
Fantle-Lepczyk Jean 5
Angulo Elena 2 6
Bang Alok 7
Dobigny Gauthier 1 8
Courchamp Franck 2
1 CBGP, Univ Montpellier, CIRAD, INRAE, Institut Agro, IRD , Montferrier-sur-Lez , France
2 Université Paris-Saclay, CNRS, AgroParisTech, Ecologie Systématique Evolution , Orsay , France
3 Institute for Global Food Security, School of Biological Sciences, Queen’s University Belfast , Belfast , United Kingdom
4 School of Biological Sciences, King’s College, University of Aberdeen , Aberdeen , United Kingdom
5 School of Forestry & Wildlife Sciences, Auburn University , Auburn , AL , USA
6 Estación Biológica de Doñana (CSIC) , Sevilla , Spain
7 Society for Ecology Evolution and Development , Wardha , India
8 Unité Peste, Institut Pasteur de Madagascar, BP 1274 Ambatofotsikely Avaradoha , 101 Antananarivo , Madagascar
Mahmood Haider
Electronic publication date: 2023 Mar 24
Publication date: 2023
Volume: 11
Electronic Location ID: e14935
Received 2022 Jul 13; Accepted 2023 Jan 31
Copyright: ©2023 Diagne et al.
Copyright year: 2023
Copyright holder: Diagne et al.
License: This is an open access article distributed under the terms of the Creative Commons Attribution License, which permits unrestricted use, distribution, reproduction and adaptation in any medium and for any purpose provided that it is properly attributed. For attribution, the original author(s), title, publication source (PeerJ) and either DOI or URL of the article must be cited.
License URL: https://creativecommons.org/licenses/by/4.0/

Keywords: Damage costs, InvaCost, Management expenditures, Monetary impact, Rodents, Reporting bias

Funding: The French National Research Agency ANR-14-CE02-0021 The BNP-Paribas Foundation Climate Initiative for funding the Invacost project The AXA Research Fund Chair of Invasion Biology The 2017-2018 Belmont Forum and BiodivERsA joint call for research proposals, under the BiodivScen ERA-Net COFUND programme AXA Research Fund Chair of Invasion Biology of University Paris Saclay The BiodivERsA-Belmont Forum Project “Alien Scenarios” BMBF/PT DLR 01LC1807C The Leverhulme Trust ECF-2021-001 The European Union’s Horizon 2020 research and innovation programme Marie Skłodowska-Curie fellowship 747120 Auburn University School of Forestry and Wildlife Sciences This work was supported by the French National Research Agency (ANR-14-CE02-0021) and the BNP-Paribas Foundation Climate Initiative for funding the Invacost project which allowed the construction of the InvaCost database. This work was initiated following a workshop funded by the AXA Research Fund Chair of Invasion Biology. This research was also funded through the 2017-2018 Belmont Forum and BiodivERsA joint call for research proposals, under the BiodivScen ERA-Net COFUND programme. Funds for Elena Angulo and Liliana Ballesteros-Mejia came from the AXA Research Fund Chair of Invasion Biology of University Paris Saclay. Christophe Diagne was funded by the BiodivERsA-Belmont Forum Project “Alien Scenarios” (BMBF/PT DLR 01LC1807C). Ross N. Cuthbert received funding from the Leverhulme Trust (ECF-2021-001). Thomas W. Bodey received funding from the European Union’s Horizon 2020 research and innovation programme Marie Skłodowska-Curie fellowship (Grant No. 747120). Jean Fantle-Lepczyk received travel support to attend the Invacost workshop from Auburn University School of Forestry and Wildlife Sciences. The funders had no role in study design, data collection and analysis, decision to publish, or preparation of the manuscript.

==============================
Background

Rodents are among the most notorious invasive alien species worldwide. These invaders have substantially impacted native ecosystems, food production and storage, local infrastructures, human health and well-being. However, the lack of standardized and understandable estimation of their impacts is a serious barrier to raising societal awareness, and hampers effective management interventions at relevant scales.

Methods

Here, we assessed the economic costs of invasive alien rodents globally in order to help overcome these obstacles. For this purpose, we combined and analysed economic cost data from the InvaCost database—the most up-to-date and comprehensive synthesis of reported invasion costs—and specific complementary searches within and beyond the published literature.

Results

Our conservative analysis showed that reported costs of rodent invasions reached a conservative total of US$ 3.6 billion between 1930 and 2022 (annually US$ 87.5 million between 1980 and 2022), and were significantly increasing through time. The highest cost reported was for muskrat Ondatra zibethicus (US$ 377.5 million), then unspecified Rattus spp. (US$ 327.8 million), followed by Rattus norvegicus specifically (US$ 156.6 million) and Castor canadensis (US$ 150.4 million). Of the total costs, 87% were damage-related, principally impacting agriculture and predominantly reported in Asia (60%), Europe (19%) and North America (9%). Our study evidenced obvious cost underreporting with only 99 documents gathered globally, clear taxonomic gaps, reliability issues for cost assessment, and skewed breakdowns of costs among regions, sectors and contexts. As a consequence, these reported costs represent only a very small fraction of the expected true cost of rodent invasions (e.g., using a less conservative analytic approach would have led to a global amount more than 80-times higher than estimated here).

Conclusions

These findings strongly suggest that available information represents a substantial underestimation of the global costs incurred. We offer recommendations for improving estimates of costs to fill these knowledge gaps including: systematic distinction between native and invasive rodents’ impacts; monetizing indirect impacts on human health; and greater integrative and concerted research effort between scientists and stakeholders. Finally, we discuss why and how this approach will stimulate and provide support for proactive and sustainable management strategies in the context of alien rodent invasions, for which biosecurity measures should be amplified globally.

Introduction

Rodents—the most abundant and diverse order of living mammals (∼40% of mammalian biodiversity; Burgin et al., 2018)—are undoubtedly the most common non-domestic vertebrates to accompany humans in their global dispersal (Cucchi et al., 2020). The ever-increasing anthropization of natural habitats coupled with many rodents’ ecological plasticity, has resulted in the continuous spread of numerous non-native rodents worldwide (Dalecky et al., 2015; Hima et al., 2019; Mazza et al., 2020; Hassell et al., 2021). Once established, these invasive alien rodents (hereafter ‘invasive rodents’) usually represent a multisectoral threat to local biodiversity (Sainsbury et al., 2020), public health (Han et al., 2015; Meerburg, Singleton & Kijlstra, 2009a), human well-being (Colombe et al., 2019) and socio-economic activities (Murray et al., 2018). We note that this definition of invasive rodents does not include indigenous rodent populations that may attain pest status and themselves have socio-environmental impacts following intermittent outbreaks and/or range expansion in their native areas (e.g., Mastomys natalensis in Tanzania; Mwanjabe, Sirima & Lusingu, 2002). 

Invasive rodents have numerous detrimental impacts on invaded ecosystems, resulting from both direct (e.g., competition, predation, destruction through digging and gnawing) and indirect (e.g., transmission of diseases, reductions in pollination efficiency or nutrient recycling) mechanisms (e.g., Stokes et al., 2009; Wardle et al., 2012; Diagne et al., 2016; Russell et al., 2020). These rodents have been implicated in the decline and extinction of native biota on numerous islands worldwide (e.g., Jones et al., 2008; St Clair, 2011; Sainsbury et al., 2020). They also spread infectious diseases of major public health importance and are key reservoirs for zoonotic diseases such as plague, scrub typhus, leptospirosis and hemorrhagic fevers (Meerburg, Singleton & Kijlstra, 2009a; Han et al., 2015; Zhang et al., 2022). Furthermore, invasive rodents increase malnutrition and threaten food security through contamination, damage and consumption of food stocks and crops (Colombe et al., 2019), as well as affect economic activities and productivity (e.g., damage to seaport infrastructure and trade; Dossou et al., 2020). The sudden outbreaks of the house mouse (Mus musculus) approximately every four years in Australia —where it can result in severe crop losses over thousands of square kilometers (Singleton et al., 2005)—illustrate the pest nature of some invasive rodents. In addition, rodent infestations are perceived as a hallmark of poverty and unhealthy living conditions—even if in reality, they may also damage the goods and properties of wealthy human populations (Garba et al., 2014).

Given the multitude of ways invasive rodents affect ecosystems, it is no surprise that four rodent species (black rat Rattus rattus, house mouse M. musculus, grey squirrel Sciurus carolinensis and coypu/nutria Myocastor coypus) are listed amongst “100 of the world’s worst invasive alien species” (Lowe et al., 2000; Luque et al., 2014). Despite all these documented impacts, management efforts that efficiently and sustainably mitigate the negative effects of invasive rodents remain limited in scope, patchily distributed and/or largely restricted to post-establishment actions (e.g., control or eradication campaigns in insular areas; Duron, Shiels & Vidal, 2017). Even then, such actions can be impaired by natural or anthropogenic reinvasions by the targeted rodent species (e.g., Harris et al., 2012). Efficient and trans-boundary efforts to prevent or limit rodent invasions in a sustainable way are urgently needed, but have remained unpopular with decision makers. An improved understanding of the impacts associated with biological invasions, and how human society contributes to them, could help to motivate greater investment to reduce economic impacts (Courchamp et al., 2017; Latombe et al., 2017; Bacher et al., 2018; Diagne et al., 2021a; Diagne et al., 2021b).

In this context, relying on monetized impacts of invaders appears as a relevant option for raising public awareness and helping to set cost-effective and sustainable management programmes (Diagne et al., 2020a; Gruber et al., 2021). Investigating costs coming from damage (economic losses due to direct and indirect impacts) and management (monetary expenditures to prevent and/or mitigate these impacts) is particularly relevant for rodents, which are, for example, responsible for massive annual loss estimates in Asia (US $1.9 billion; Nghiem et al., 2013), Tanzania (US$ 45 million; Leirs, 2003), United States of America (US$ 19 billion; Pimentel, Zuniga & Morrison, 2005) and Australia (US$ 60 million; (Brown & Singleton, 2000)). However, a global overview still remains necessary for the purpose of both research (e.g., identifying gaps and priorities; Diagne et al., 2020a) and management (e.g., coordinating regional biosecurity measures, particularly for areas with restricted capacities; Early et al., 2016). Here, we provide such a global synthesis of the reported economic costs of invasive rodents, by synthetizing and investigating how these costs are distributed across taxa, geographic areas and socio-economic sectors over time. From there, we highlight crucial knowledge gaps, identify further research perspectives and propose cost-based recommendations for efficient management of rodent bioinvasions.

Materials & Methods

Data collection and processing

We used a four step procedure to collate and process the global economic cost data of invasive rodents (Fig. 1). First, (Fig. 1A; Appendix 1, ‘Rodentia dataset’ tab), we selected cost entries identified as Rodentia in the ‘Order’ column of the most recent version (at the time of writing) of the InvaCost database (version 4.1, available at https://doi.org/10.6084/m9.figshare.12668570, which includes 13,553 cost entries collated from scientific and grey literature in multiple languages; Angulo et al., 2021; Diagne et al., 2020b). Each cost entry recorded is standardized to 2017 US dollars and categorized by a range of 65 descriptive fields (Appendix 1, ‘Descriptors’ tab). We added new data by contacting appropriate experts and agencies working on rodent invasions to seek cost information, and scanning these novel references to discover additional publications or reports using a ‘snowball’ process. Every new cost record was integrated following the InvaCost template and added to the original Rodentia dataset, so that we obtained our final starting Rodentia dataset (Fig. 1A; Appendix 1, ‘Rodentia dataset’ tab).

Figure 1 Workflow depicting the data collection and filtering process.

The expanded subset was obtained through the ‘expansion’ of the suitable subset using the ‘invacost’ R package (Leroy et al., 2022). The criteria used for generating the conservative subset were based on the ‘Implementation’ (observed versus potential costs) and ‘Method_reliability_final’ (high versus low-reliability costs) fields of the InvaCost database (Appendix 1, ‘Descriptors’ tab). The number of taxa includes both individual species and undefined species aggregated at the genus level. Costs are expressed in 2017 US$.

Second, we carefully checked the data to (i) remove overlapping or duplicated costs, (ii) assess the reliability (high or low) of the estimation approach used to provide each cost figure based on evaluation criteria similar to those considered by Bradshaw et al. (2016) and (iii) remove all cost entries without clear information on their duration, calculated as the number of years between the recorded cost entry’s starting (‘Probable starting year adjusted’ column) and ending (‘Probable ending year adjusted’ column) years (Fig. 1B). All modifications and additions made here were synthesized in Appendix 1 (‘Changes made’ tab) and systematically sent to updates@invacost.fr as recommended by the database managers.

Third, the resulting subset (Appendix 1, ‘Suitable subset’ tab) was homogenized using the expandYearlyCosts function of the ‘invacost’ R package (Leroy et al., 2022) so that all cost entries were considered on an annual basis (hereafter ‘annualized cost entries’)—which means that costs spanning multiple years were divided according to their duration time (e.g., $20 million between 1991 and 2000 becomes $2 million annually across those years). While this cost breakdown over time is reasonable to obtain comparable cost estimates and allow further, relevant estimations, it is unlikely to be an accurate reflection of the actual cost development within a single action over time. However, this annual cost information is often missing from the source documents providing the total cost, which instead report large sums over multi-year periods.

Finally, the resulting subset with annualized cost entries (Fig. 1C; Appendix 1, ‘Expanded subset’ tab) was filtered using two successive filters to obtain our final subset (Fig. 1D; Appendix 1, ‘Conservative subset’ tab): (1) we kept only observed costs by using the ‘Implementation’ column to exclude any potential (i.e., predicted but not incurred costs); (2) we retained only high-reliability costs by using the ‘Method reliability’ and ‘Method reliability refined’ columns (with the latter, if provided, favored over the former in case of non-congruent information) to exclude costs without documented and repeatable methodologies. Our final, conservative subset contained 609 annualized cost entries between 1930 and 2022 (Fig. 1, Appendix 1, ‘Conservative subset’ tab). This conservative subset, unless otherwise stated, was considered for further analyses below.

Temporal dynamics of costs

We examined how costs developed over time by applying the summarizeCosts function of the ‘invacost’ R package (Leroy et al., 2022) to our conservative subset. This function provides the observed cumulative and average costs over a specific period of time (here, the whole period covered by our conservative subset) and at different time intervals (here, 10-year intervals), which allows us to display the temporal trend of invasion costs over time. The cumulative costs incurred were calculated as the sums of all cost estimates provided in the ‘Cost_estimate_per_year_2017_USD_exchange_rate’ column of the conservative subset (Appendix 1), and the average cost amount for each decade by dividing the cumulative cost by ten years.

Taxonomic representativeness

To evaluate the proportion of invasive rodent species for which cost data were available within each taxonomic family, we compared the list of individual rodent species reported in the ‘Rodentia subset’ with comprehensive lists of invasive rodents recorded worldwide, following an approach similar to Cuthbert et al. (2021b). Lists of known invasive rodents were extracted and compiled from the Global Invasive Species Database (GISD; http://www.iucngisd.org/gisd/) and the sTwist database (version 2; Seebens et al., 2020). We filtered these databases to select only species belonging to the order Rodentia, using the Backbone Taxonomy from the Global Biodiversity Information facility (https://www.gbif.org/) to standardize species names and remove any duplicates. For the first records (sTwist) database, we selected only those exotic taxa that were known to be presently established. We classified all such species as invasive, but note that the definitions of invasiveness may differ slightly between these datasets (Cuthbert et al., 2021b).

Cost calculation and distribution

The total cost for each category (see below) was obtained by summing all annualized cost estimates provided in the ‘Cost_estimate_per_year_2017_USD_exchange_rate’ column of our conservative subset. Again, total costs were obtained by summing all annualized cost entries (‘Cost_estimate_per_year_2017_USD_exchange_rate’ column). Using key database descriptors (see Appendix 1, ‘Descriptors’ tab for details on all variables and categories used here), we subsequently investigated the breakdown of cost data and estimates across:

(i) Taxa: considering the ‘Species’ descriptor; undetermined species were therefore aggregated by genus, where possible (e.g., undefined Rattus sp. and Rattus spp. were grouped under the single category ‘Rattus spp.’);

(ii) Geography: considering the ‘Geographic region’ descriptor, with Central America merged with North America and Pacific Islands merged with Oceania; we also considered the insular habitat status (yes or no) using the proposed ‘Island2′descriptor;

(iii) Type of cost: Damage (economic losses due to direct and indirect impacts of rodents) versus Management (monetary investments to prevent and/or mitigate impacts—further separated according to type of actions undertaken (pre-invasion management, post-invasion management, knowledge funding and mixed management)) and;

(iv) The impacted sector: (Agriculture, Authorities-Stakeholders, Environment, Fishery, Forestry, Health, Public and social welfare).

For each descriptor, we grouped all cost entries that were not unambiguously assigned to one of the above-mentioned specific categories under the category mixed.

Results

Global costs and temporal dynamics

Our analyses revealed that invasive rodents have already cost the global economy at least US$ 3.6 billion (annually US$ 38.7 million) between 1930 and 2022, based on the cost estimates reported between 1930 and 2022. This average estimate was increased until US$ 87.5 million annually when considering the timescale 1980-2020 –i.e., the period that concentrated most (∼97%) of the data recorded in the conservative subset (Fig. 2; Appendix 1). A less conservative approach (i.e., using also low- reliability and potential cost data as well) produced a global figure of around US$ 297.4 billion worldwide for the period 1930–2022 (Fig. 1C, Appendix 2). The dynamics of costs showed an exponential increase over time (Fig. 2), whatever the nature of cost data considered–while an artifactual decrease can be observed for recent years due to the multi-year delays between the occurrence and reporting of costs in the literature (Appendix 3). All cost figures shown in this section derived from the conservative subset are summarized in Appendix 4.

Figure 2 Temporal trend of global rodent invasion costs (in millions of 2017 US$) between 1980 and 2020.

The solid line represents the temporal dynamics of costs based on a linear regression, while the dashed line connects the average annual costs for each decade (see Leroy et al., 2022 for methodological details). The horizontal bars indicate the total time span over which decadal mean costs were calculated.

Taxonomic representativeness and distribution of costs

Invasion costs were reported for 15 individual rodent species in our conservative data subset, but there are at least 49 invasive alien rodents recorded worldwide (i.e., across InvaCost, sTwist and GISD; Fig. 3). Two further species recorded in the original InvaCost database were not included in our conservative subset (Fig. 3). Specifically, costs for Hystrix brachyura and Sciurus niger either reported (for H. brachyura in the UK) or expected (for S. niger, should it arrive in the Netherlands), were respectively deemed as low-reliability and potential estimates and thus conservatively excluded. The most underrepresented rodent families in our subset include Sciuridae (11 species without costs out of 18), Muridae (nine species out of 13) and Cricetidae (five species out of seven) (Fig. 3).

Figure 3 Taxonomic bias in the costs of invasive rodents.

Invasive rodent species are those recorded in the InvaCost database, the Global Invasive Species Database (GISD; http://www.iucngisd.org/gisd/) and the sTwist database (version 2; Seebens et al., 2020). Species with reported costs are in green rolls, while species without reported costs are in yellow rolls, all grouped following their taxonomic family. Species with dichromatic rolls (H. brachyura, S. niger) were in the original Rodentia subset, but were not considered in our conservative subset. Roll height is scaled to the number of species within each group and species silhouettes are sized to scale. Grey bars show total cumulative costs in 2017 US$ (log10 scale). Except Glis glis, Histrix brachyura, and Tamias sibiricus that were created by the authors, all animal silhouettes were obtained from an open source platform (http://phylopic.org/) where the silhouette of Rattus norvegicus was created by Rebecca Groom.

Costs were skewed towards the muskrat Ondatra zibethicus (US$ 377.5 million (4.1 million/year); n = 18 annualized cost entries), undefined rats Rattus spp. (US$ 327.8 million (3.5 million/year); n = 96), the brown rat R. norvegicus (US$ 156.6 million (1.7 million/year); n = 66) and the North American beaver Castor canadensis (US$ 150.4 million (1.6 million/year); n = 32). These four taxa constituted about a third of the total costs reported. All remaining species-specific costs totaled less than US$ 100 million, but mixed costs from diverse or unspecified taxa collectively amounted to US$ 2.4 billion (25.4 million/year). Despite being the species with the highest number of annualized entries (n = 117), costs from the coypu M. coypus totaled “only” US$ 90.9 million.

Considering only damage costs, O. zibethicus (US$ 328.6 million (3.5 million/year); n = 8) was the costliest species, followed by R. norvegicus (US$ 68.6 million (0.7 million/year); n = 8) and C. canadensis (US$ 65.4 million (0.7 million/year); n = 10). The only specific species with more than 10 damage cost entries were M. coypus (n = 70), Callosciurus erythraeus (n = 32) and Sciurus aureogaster (n = 12), which totaled US$ 64.7 million, US$ 1.9 million and US$ 19.8 million, respectively.

Conversely, management costs were mostly associated with C. canadensis (US$ 84.9 million; n = 22), R. norvegicus (US$ 79.2 million; n = 57) and O. zibethicus (US$ 48.8 million; n = 10). While the aggregated Rattus spp. group incurred the second highest damage costs (US$ 304.2 million; n = 4), they represented only the sixth highest spend for management actions (US$ 23.4 million; n = 89).

Cost distribution across types, space and sectors

Most costs (87%) corresponded to damages or losses (US$ 3.1 billion (33.7 million/year), n = 162) despite a lower number of reported estimates when compared with management expenditures (n = 426). Spending related to the latter was dominated by post-invasion management (US$ 381.2 million (4.1 million/year), n = 314), which was 50-times greater than pre-invasion management (Fig. 4A).

Figure 4 Cost distributions across species according to (A) cost type, (B) geographic region and (C) impacted sector.

The size of each node corresponds to the cost total calculated based on the data reported in the conservative subset. Total cumulative costs are expressed in 2017 US$ millions.

Regionally, most costs were incurred in Asia (60%; US$ 2.2 billion, n = 109), followed by Europe (19%; US$ 678.0 million, n = 275), North America (9%; US$ 329.9 million, n = 74), South America (6%, US$ 211.3 million, n = 48) and Oceania-Pacific Islands (6%, US$ 204.6 million, n = 89), with remaining regions contributing US$ 1 million or less each (Fig. 4B). Many species had recorded impacts in only a few geographic regions as a result of restricted invasive ranges. For example, O. zibethicus only incurred costs in Europe, the single continent invaded by this rodent species. Islands suffered from higher total reported costs than mainlands overall (US$ 284.4 million, n = 254 versus US$ 129.1 million, n = 158) (Fig. 5), with the vast majority of reported costs on mainlands (US$ 96.9 million, n = 35) being damage-related, while about two thirds of the total costs reported from islands were for management (US$ 199.2 million, n = 234). Post-invasion actions dominated management spending overall, for both islands and mainlands (Fig. 5).

Figure 5 Cost estimates (log10 scale) according to the type of damage or management expenditures between mainland (in green) and island (in orange) areas. Pre-invasion management: monetary investments for preventing successful invasions.

Pre-invasion management: monetary investments for preventing successful invasions in an area (including quarantine or border inspection, risk analyses, biosecurity management, etc.); post-invasion management: money spent for managing invasions in invaded areas (including control, eradication, containment); knowledge funding: money allocated to all actions and operations that could be of interest at all steps of management at pre- and post-invasion stages (including administration, communication, education, research); or mixed: costs that included at least (and without possibility to disentangle the specific proportion of) two of the previous categories; management/damage: costs that included both cost types. Total cumulative costs are expressed in 2017 US$ (log10 scale).

Regarding impacted sectors, most costs were incurred by the Agricultural sector (63%; US$ 2.3 billion; n = 110) with 93% of this cost recorded in Asia, followed by expenditures by authorities and stakeholders (26%; US$ 928.3 million; n = 447), of which slightly less than half occurred in Europe (Fig. 4C). However, almost all (95%) of the agricultural costs were attributed to mixed taxa, and for the costliest individual species, O. zibethicus (n = 18), 91% of the costs were borne by Authorities and stakeholders.

Discussion

Tremendous, increasing and uneven economic costs

Invasive rodents have conservatively cost the global economy at least US$ 3.6 billion reported between 1930 and 2022, representing an average annual cost of US$ 87.5 million in the period 1980-2020 (where most data have been reported). Inclusion of all costs through a less conservative data filtering leads to a global amount more than 80-times higher (US$ 297.4 billion; Fig. 1). Importantly, all cost figures shown here should be considered as orders of magnitude rather than exact cost estimates, given the clearly non-exhaustive representation and evolving nature of the cost data considered. Nevertheless, the costs show an undeniable increase over time. While it has been recently shown that costs of biological invasions are rising globally even after accounting for research effort (Haubrock et al., 2022), disentangling the extent of the effects of ‘increased cost reporting’ from those of ‘actual increase in cost amounts’ remains challenging (Diagne et al., 2021a). Applying a range of modelling techniques to our data (Appendix 5, Appendix 6; Leroy et al., 2022)—which allow us to take into account the statistical issues typical to econometric data (e.g., heteroskedasticity, temporal autocorrelation and outliers) as well as potential time lags between cost occurrence and their reporting—(i) provides support for this increasing trend over time and (ii) illustratively leads to a cost estimate that could reach US$ 7.6 billion for the single year 2020. The latter, for instance, is a value exceeding the European Union’s negotiated budget for addressing the COVID-19 crisis (US$ 7.3 billion, consilium.europa.eu). Nevertheless, the global cost figure displayed here (US$ 3.6 billion) is unevenly distributed across taxa, space, sectors and types of costs. Although caution should be exercised in interpretation (see ‘An undervalued economic burden’ below), this breakdown evidenced some interesting insights that highlight current research biases.

From a taxonomic perspective, our study further supports the “major threat” status of multiple rodent invaders (Howald et al., 2007, Cuthbert et al., 2021a; Diagne et al., 2021a). We found that a significant proportion of costs attributable to a refined taxonomic level (US$ 500.4 million; 14% of the total cost) were caused by species within the genus Rattus. This genus is well-recognized as containing a number of highly impactful invaders worldwide (Lowe et al., 2000; Luque et al., 2014; Cuthbert et al., 2021a), with documented multi-sectoral impacts including a role as disease reservoirs (Morand et al., 2015), reductions and alterations in socio-economic activities (Murray et al., 2018) and negative impacts on biodiversity and ecosystems (Doherty et al., 2016). However, as a result there is likely more intensive research effort—and thus likely more cost information—on these species (Zeng et al., 2018) compared with other rodent taxa. Similarly, O. zibethicus, the species with the highest reported costs, has been officially ranked in the European list of the most concerning invasive alien species (https://ec.europa.eu/environment/nature/invasivealien/list/index_en.htm).

From a geographic perspective, the higher reporting rates observed in Europe, North America and Oceania most probably reflect skewed research efforts and/or economic capacities rather than a true spatial distribution of costs, as shown for invasion science in general (Bellard & Jeschke, 2016; Nuñez et al., 2022). For instance, costs of invasive rodents in Africa represented less than 1% of the total global reported cost; yet common invasive rodents (R. rattus, R. norvegicus and M. musculus) are known to have similar impacts there as in the rest of the world (Dossou et al., 2020; van Wilgen et al., 2020). Nonetheless, Asia somewhat surprisingly comprised the highest proportion of the total costs, although this was mainly due to a single value associated with agricultural losses from Mus and Rattus species in Malaysia, Myanmar and Thailand (Nghiem et al., 2013), highlighting the crucial importance of each additional case study. Furthermore, while damage costs comprised the majority of mainland costs, management expenditure was more common on islands. This likely reflects high local conservation efforts (particularly investments in preventative measures and eradication campaigns) in these fragile/threatened insular ecosystems, which support disproportionately high levels of native species endemism, with high risk of local extinction often at least partly linked to invasive rodents (Bellard et al., 2017).

From a sectoral perspective, our results illustrate the intrinsic multi-sectoral nature of invasive rodent impacts (Colombe et al., 2019). For example, a single invasive rodent, the Eastern grey squirrel S. carolinensis, may simultaneously impact local biodiversity, spread zoonotic pathogens, consume ornamental plants and cause tree damage by bark stripping activity (e.g., Millins et al., 2015; Broughton, 2020). Considering these impacted sectors separately, agricultural losses unsurprisingly comprised the greatest proportion, with other sectors having lower monetized impacts. This pattern could be explained in at least two ways. On the one hand, rodents are among humans’ most important competitors for food resources globally (Meerburg, Singleton & Leirs, 2009b; John, 2014; Belmain et al., 2015), which inevitably leads to massive production losses while stimulating high financial management efforts to mitigate such impacts. On the other hand, accurately quantifying monetary losses in non-commercial sectors such as public health is not so straightforward (Diagne et al., 2021a), which contributes to explain why agriculture-based costs are more (easily) evaluated and thus reported.

An undervalued economic burden

The high costs evidenced here still obviously represent a massive underestimation of the true costs globally incurred by invasive rodents. Indeed, we first decided to conservatively examine only the data subset deemed to be the most robust (Fig. 1). This exclusion of unsubstantiated costs (e.g., those relying on unsourced or unclear calculations) also contributes to explain the striking discrepancy between our resulting total cost and cost estimates from previous works (e.g., US$ 19 billion per year only for the United States; Pimentel, Zuniga & Morrison, 2005).

Second, our synthesis is not exhaustive because of methodological, logistical and cost-intrinsic factors already evoked more broadly in the context of global invasive alien species burdens (Diagne et al., 2021a). Notably, the accessibility of grey literature materials varies, in particular in non-English materials (Angulo et al., 2021), the monetary valuation of non-market ecosystem services is not straightforward (Kallis, Gómez-Baggethun & Zografos, 2013) and there is active ethical debate on the monetization processes (Meinard, Dereniowska & Gharbi, 2016). An illustrative case of missing, yet potentially massive, costs can be found in the scarce management expenditure reported here in mainland areas. This apparent lack of management costs in these areas rather provides support for unreported, but probably extant cost information from numerous local pest control organizations (PCOs), which have adopted preventive and control strategies to limit rat and mouse populations within urban settlements (Maas et al., 2020). In addition, millions of inhabitants suffer from unevaluated rodent damages (e.g., Garba et al., 2014) and invest into sometimes informally traded rodenticides every day (Dalecky et al., submitted), with many of the rodent species targeted being invasive, especially within cities. However, the associated costs were not found through our search protocols, and are not necessarily matters of public record.

Third, about two-thirds of known invasive rodents had no invasion costs reported (Fig. 3), yet it is unlikely they have had no economic burden given their impacts on socio-ecosystems. Even for species with some recorded costs, the discrepancy between the known, ubiquitous impact and low number of records shows the enormous gap between observed costs and potential costs. For example, this is the case for the house mouse, for which massive impacts are recorded worldwide.

Fourth, there is a recurrent lack of distinction between invasive and native rodent ‘pest’ species in reports. The costs reported in these documents had to be disregarded in our analysis because the proportion of the impact actually due to the invasive rodents could not be accurately ascertained. The commensal habits and long-standing invasion history of some rodent species, particularly mice and rats, make them very often classified as generic ‘pests’ rather than specifically as invasive species (Stenseth et al., 2003), and so they are treated indifferently within the literature, especially outside ecology (e.g., agriculture and health). In this instance, the non-specific search terms used within InvaCost may be not optimally designed for capturing such information, and as a result a large part of the actual costs may be missed. Lastly, impacts of invasive rodents on human well-being are often quantified but not monetized (Diagne et al., 2021a). For instance, invasive rats were estimated to consume food crops that could feed 200 million people in Asia for an entire year (Singleton, 2003), and it was estimated that 280 million cases of undernourishment could be avoided worldwide through proactive rodent control (Meerburg, Singleton & Leirs, 2009b). Similarly, wild populations of the house mouse periodically undergo severe outbreaks, which cause substantial damage to cropping landscapes in South-Eastern Australia (Brown et al., 2022). However, even in these extreme cases, cost estimations are scarce or simply missing. Rodents (including most invasive ones) are also major reservoirs of zoonotic diseases responsible for over 400 million human illness cases each year (Meerburg, Singleton & Kijlstra, 2009a; Colombe et al., 2019)—which, in addition to non-monetizable injuries and deaths, lead to a cascade of socio-economic impacts, presumably with associated costs (e.g., diseases surveillance and control; diagnostics and medical care, Disability Adjusted Life Years (DALY) and decreased productivity and education). As an illustration, invasive R. norvegicus plays a pivotal role in the epidemiological cycle of leptospirosis in many urban settings, responsible for a global loss of 2.9 million DALY annually, corresponding to approximatively one million cases (Torgerson et al., 2015). In the hypothetical scenario where only half of these cases are treated at only US$ 10 per person, medical care (which is only one facet of the overall cost burden) for this one scenario alone would already reach US$ 5 million annually. Furthermore, these health-associated costs are expected to dramatically increase as ongoing land use change and urbanization amplify the role of (invasive) rodents as zoonotic reservoirs in many locations (Gibb et al., 2020; Mendoza et al., 2020; Hassell et al., 2021).

A call for concerted research and management efforts

Improving cost estimation and reporting

We stress the need for more accurate and standardized multiscale cost estimations towards currently under-reported regions, taxa and sectors. In addition to the recommendations already made by Diagne et al. (2021a), Diagne et al. (2021b) and Cuthbert et al. (2022) for better cost data reporting, we advocate here for further improving cost estimation of invasive rodents in at least two ways.

On the one hand, we highlight the need to disentangle species-specific contributions to the costs reported, which helps to set priorities in local contexts and evaluate cost-effectiveness of management actions that may differ at the species level (Gruber et al., 2021). Indeed, 67% of the total costs estimated here were associated with mixed invasive rodent species. At least, we strongly encourage separation of invasive versus native status of rodent species in impact assessments, rather than considering all species only under a “pests versus non-pests” dichotomy. This increased granularity across scales will enhance our understanding of rodent impacts, and improve the targeting and effectiveness of communication campaigns and management actions (Diagne et al., 2020a; Gruber et al., 2021).

On the other hand, standardized and multilingual cost surveys should be tailored and distributed across an identified set of stakeholders facing or dealing with rodent invasions (e.g., farmers, pest control agencies, ports and safety services), and pertaining to a variety of impacted sectors (e.g., health, agriculture, transports). Indeed, cost data likely exists from these stakeholders but a lack of capacity, time or interest often hampers making this cost information readily or publicly available. In addition, obvious language barriers for capturing existing data still remain, which also likely contributed to the data unevenness presented in this study (but see Angulo et al., 2021). Therefore, we strongly encourage efforts that will engage in gathering these harder-to-access data through the survey proposed above. As an illustration of the usefulness of such an approach, 50 of the 136 invasive rat control projects identified in Duron, Shiels & Vidal (2017)’s inventory came from responses to a questionnaire (written in English and French) circulated through invasion and conservation web lists (e.g., Aliens-L) and personal networks. This illustrates the urgent need for long-standing partnerships among expert scientists, governmental and non-governmental agencies and local stakeholders. We believe the InvaCost consortium (http://invacost.fr/en/accueil/) could serve as a foundation for such a network, in which other existing global efforts to compile information—such as the Database of Island Invasive Species Eradications (DIISE; Spatz et al., 2022) or the Global Register of Introduced and Invasive Species (GRIIS; Pagad et al., 2018)—should be valuably integrated.

Operationalizing cost-based research

Accurate knowledge and consistent accounting of the economic costs is integral to coordinated, efficient and sustainable management of rodent invasions. First, local cost estimates are essential to raise awareness and thus incentivize and facilitate community buy-in to subsequent control and prevention programs. Indeed, communicating the magnitude of these impacts in an accessible way is critical to creating a supportive legislative, political and societal environment, which is a basis for long-term policies on rodent invasions (Novoa et al., 2017; Adamjy et al., 2020). We advocate for relying on this cost-based information to stimulate more efforts from decision makers towards implementing biosecurity measures (i.e., prevention, detection, and rapid response), which represent the most efficient and cost-effective approaches to limit invasions and their impacts (Matos et al., 2018; Cuthbert et al., 2022).

Second, relying on a common, standardized metric (i.e., currencies) to quantify impacts of invasions allows for consistent monitoring and comparison over time and across regions. In turn, this facilitates the assessment of efficiency (e.g., cost-effectiveness analysis; McConnachie et al., 2016), prioritization (e.g., in addition to qualitative indicators of invasions’ impacts; Evans et al., 2018) and expenditure balance (e.g., in the case of regional biosecurity measures; Faulkner, Robertson & Wilson, 2020) of management actions. Lastly, considering economic costs would help to design locally adapted, and thus sustainable, management strategies that account for economic and societal realities, as seen in the successful co-construction and implementation of ecologically-based rodent management (EBRM) strategies with local communities (Constant et al., 2020). This is particularly critical in low- and middle-income countries, where economic resources are scarce, and societal concerns are mostly dominated by food and health security (Crowley, Hinchliffe & McDonald, 2017; Evans et al., 2018). Therefore, we encourage increasing management efforts through closer science-society interactions (Novoa et al., 2018).

Conclusion

Whether they are long term human commensals (rats, mice), invaders of specific habitats (beavers, muskrats, coypus) or newly invasive species from exotic pet trades (squirrels, dormice), invasive rodents are particularly ubiquitous yet individually relatively inconspicuous. We show here that even the small fraction of their impacts that have been monetized is sufficient to warrant a much greater focus towards ongoing and future invasions by this taxonomic group.

Supplemental Information

Supplemental Information 1 Data considered in this study

’Descriptors’ provides full definition and details about the descriptive columns from the InvaCost database (note that we also provide definition of the new columns specifically added for our study); ‘Rodentia dataset’ contains the original cost entries pertaining to the order Rodentia from the InvaCost database (version 4.1); ‘Changes_made’ synthesizes all modifications made to the original cost data associated with rodent taxa; ‘Rodentia subset’ contains original cost entries refined as well as new cost entries collected following our own data searches (note that the latter are identified with the tag New_cost in the “InvaCost_Id” column); ‘Suitable subset’ contains the portion of cost entries from the ‘Rodentia dataset’ after filtering out overlapping or duplicating costs, as well as costs without clear temporal information; ‘Expanded subset’ contains the annualized cost entries following data expansion through the invacost R package (Leroy et al., 2022); ‘Conservative subset’ is the most robust subset of the ‘Expanded subset’ obtained after keeping only the observed (“Implementation” column) and high (“Method reliability_final” column) cost entries.

Click here for additional data file.

Supplemental Information 2 Distribution of cost entries and estimates recorded in the Rodentia subset according to their reliability (high versus low) and their implementation (potential versus observed)

All details on the descriptive fields considered are available in Appendix 1.

Click here for additional data file.

Supplemental Information 3 Temporal trends of the cost estimates of invasive alien rodents from our study

We considered (a) the expanded subset and (b) the conservative subset (see Fig. 1 and Appendix 1 for further details on the subset and filtering steps). In (a), the trend is described separately for potential and observed cost entries (see “Implementation” column; Appendix 1). In (b), trends are described separately for damage, management and mixed costs (see “Type of cost merged” column; Appendix 1). Costs are provided in millions of 2017 US$. The horizontal dotted lines represent annual averages over the decadal time period, solid bars represent 10-year means and filled circles represent annual costs scaled by size to match the number of entries.

Click here for additional data file.

Supplemental Information 4 Summary of the cost distribution per invasive alien rodent taxon, impacted sector, geographic region and type of costs from the conservative subset

For each line, the ‘Total_cost_estimate’ is calculated as the sum of cost values provided in the ‘Cost_estimate_per_year_2017_USD_exchange_rate’ column of the conservative subset. The number of annualized cost entries is provided in parenthesis. All details on the descriptive fields considered are provided in Appendix 1.

Click here for additional data file.

Supplemental Information 5 Temporal trends in rodent invasion costs considering a range of statistical models

Ordinary least squares (OLS), robust regression, generalised additive model (GAM), multivariate adaptive regression splines (MARS) and quantile regressions (0.1 (lower boundary of cost), 0.5 (median cost value), 0.9 (upper boundary of cost)). Shaded areas are 95% confidence intervals (except MARS, which are prediction intervals); points represent annual totals. The y-axes are on a log10 scale, and are scaled separately among subplots. Models are generated from the modelCosts function of the ‘invacost’ package—note that we removed cost data after 2011 for model calibration due to the time lags in cost reporting that may result in an artifactual decrease of cost over the most recent years (see Leroy et al., 2022 for all methodological details). Model evaluation was based on the assessment of their predictive performance via root-mean-square deviation (RMSE) and the level of variance explained. Models varied in their goodness of fit (RMSE 0.60–0.95; Appendix 6), with costs for 2020 projected between US$ 21.9 million (linear robust regression) and US$ 7.6 billion (MARS). Quantiles were increasingly divergent through time, indicating greater amplitudes between lower and upper bounds in recent years. Although predictions inherently vary among models, consistent outcomes across approaches offers strong support for the resulting temporal trends, and a robust method through which to assess the general tendency of all models rather than attempting to identify a putative best model across sometimes sparse data.

Click here for additional data file.

Supplemental Information 6 Estimates (2017 US$) and root mean square errors (RMSE) for models in Appendix 5

Ordinary least squares (OLS), robust regression (RR), multivariate adaptive regression splines (MARS) and generalised additive model (GAM), as well as quantile regressions. Models considered annual total invasion costs as a function of time, between 1930 and 2020.

Click here for additional data file.

We are extremely grateful to the organizers of the InvaCost workshop that allowed the genesis of this project, as well as to all contacted people who kindly answered to our requests for information about the costs of invasive rodents. We thank L. Nuninger and C. Assailly for their work in the initial project, and María Angulo and Nuria Cerdá for their help in generating the Fig. 3. Last, the authors thank Dr. Steffen Oppel and another anonymous reviewer for their thorough revision of the article which greatly improved it.

Additional Information and Declarations

Competing Interests

Author Contributions

Data Availability

The authors declare there are no competing interests.

Christophe Diagne conceived and designed the experiments, performed the experiments, analyzed the data, prepared figures and/or tables, authored or reviewed drafts of the article, and approved the final draft.

Liliana Ballesteros-Mejia conceived and designed the experiments, performed the experiments, analyzed the data, prepared figures and/or tables, authored or reviewed drafts of the article, and approved the final draft.

Ross N. Cuthbert conceived and designed the experiments, performed the experiments, analyzed the data, prepared figures and/or tables, authored or reviewed drafts of the article, and approved the final draft.

Thomas W. Bodey performed the experiments, authored or reviewed drafts of the article, and approved the final draft.

Jean Fantle-Lepczyk performed the experiments, authored or reviewed drafts of the article, and approved the final draft.

Elena Angulo conceived and designed the experiments, performed the experiments, analyzed the data, prepared figures and/or tables, authored or reviewed drafts of the article, and approved the final draft.

Alok Bang performed the experiments, authored or reviewed drafts of the article, and approved the final draft.

Gauthier Dobigny performed the experiments, authored or reviewed drafts of the article, and approved the final draft.

Franck Courchamp conceived and designed the experiments, performed the experiments, authored or reviewed drafts of the article, and approved the final draft.

The following information was supplied regarding data availability:

The raw and modified data are available in the Supplemental Files.

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
