# Peer review of "Economic costs of invasive rodents worldwide: the tip of the iceberg"

_PeerJ, doi:10.7717/peerj.14935_

## Round 0.1 · original submission · Minor Revisions

Thank you very much for your nice contribution. However, reviewers suggest some minor changes before its final acceptance. So, please work on these suggestions and resubmit the paper.

·

Basic reporting

This manuscript synthesizes the reported costs associated with invasive rodents worldwide, and summarises the information by species, type, sector and geography. The authors highlight that their presented numbers will be an understatement of the true costs that are incurred globally due to underreporting, and suggest ways how cost reporting could be improved to facilitate more comprehensive assessments in the future. The paper is well and clearly written, and is based on a database and similar papers by the same author collating these costs from the literature and other sources. Overall, this is a very useful and scientifically valid manuscript that is publishable.

One suggestion for improvement would be to stress biosecurity more: the main aim of the manuscript (which I agree with and support) is to raise more societal awareness of the costs of invasive rodents (L. 23, 83-85), and in my opinion this is most critical for prevention. Unlike many other costs that are incurred AFTER the damage has been done (when it may be easier to convince people), preventative measures such as biosecurity must occur BEFORE any damage is done and are therefore incredibly difficult to fund because no politician ever sees the value in spending money on something that isn't already a massively expensive problem. I think this context could be highlighted much more in the Discussion, namely that investments in biosecurity (which are unpopular because they are considered 'expensive' and perpetual, and never have any noticeable or demonstrable outcomes, because if they work well then by definition 'nothing happens') are incredible cost-effective in that they could prevent all the massive costs that rodents may cause when invasion is not prevented. There are a number of papers highlighting the value of biosecurity to make that point:
Broome, K. 2007. Island biosecurity as a pest management tactic in New Zealand. In G. Witmer, W. Pitt, K. Fagerstone & C. Clark (eds) Proceedings of an international symposium managing vertebrate invasive species: 104–107. Lincoln: University of Nebraska.
Moore et al. 2010. Protecting islands from pest invasion: optimal allocation of biosecurity resources between quarantine and surveillance. Biol. Conserv. 143: 1068-1078.
Matos et al. 2018. Connecting island communities on a global scale: case studies in island biosecurity. Western North American Naturalist 78: 959-972.) and I think your synthesis could make a much stronger point how economic such investments could be if they prevented costs on the scale that you report.

Experimental design

There is no experimental design, as this is a review, but see my concerns over the 'design' of the time trend models below.

Validity of the findings

Most findings are valid, but there is a problem with the time trend: the authors attempt to portray the 'temporal dynamics' of costs (L. 137-151), presumably with the aim to show that they have increased massively over time. I find this problematic for three reasons:

(a) virtually everything (human population, trade volume, transport, GDP, number of research papers etc.) has increased massively over the same time horizon (1930-2020), hence the null expectation should be a corresponding increase in the reported cost of invasive rodents. I therefore do not understand why a range of modelling techniques (L. 143-149) are required in an otherwise non-technical review to state something that is unlikely to be contested (= costs increased from 1930-2020)? If you want to employ fancy models to explore the temporal dynamics of invasion costs, they should at least account for the increase in 'everything else' if you want to show that the situation is getting worse? However, I don't think such an analysis is critical for this manuscript, and may also be beyond the available data - unless the confounding factors in the data are explicitly addressed in the models.

(b) In L. 124-126 the authors state that they calculated 'annual' costs by dividing reports of longer project costs by the number of years. That is of course sensible and defensible to obtain comparable cost estimates, but it is unlikely to be an accurate reflection of the actual cost development over time. For example, the $20 million project (L. 125) between 1991 and 2000 is unlikely to have spent $2 million every year, but may have spent $1 million in 1991 and $3 million in 2000. I don't think that such 'annual averages' are of any use for the estimation of temporal trends in costs unless this 'averaging' is somehow accounted for in the models.

(c) The reported costs you have in the database are a product of what is spent and what is reported, and an increase in either of those two factors would result in an observed increase in the reported costs even if the other factor remained constant. You should acknowledge that the reporting rate may have increased, and that this may explain some of the increase in 'reported' costs - this is an incredibly difficult issue that plagues many aspects of trend analysis from citizen science data where reporting rates vary over time (e.g. Isaac et al. 2014. Statistics for citizen science: extracting signals of change from noisy ecological data. Meth. Ecol. Evol. 5: 1052-1060. Johnston et al. 2021. Analytical guidelines to increase the value of community science data: An example using eBird data to estimate species distributions. Div. Distrib. 27: 1265-1277).

On balance, I think this component of the manuscript could be removed or scaled back to a single graph/sentence stating the obvious increase in the number of reported cost estimates over time - unless the models can actually address and overcome the two main issues inherent in the data (which may well be beyond the scope of this manuscript).

Additional comments

L. 82-85: This sentence would benefit from some rewording along the lines of 'government expenditure on preventative measures is unpopular, so a better understanding of the costs associated with invasive rodents may make it easier to justify such investments'.

L. 119-121: Please clarify whether these fixes have been permanently implemented in the InvaCost database, i.e. if I were to download these data now, I would not have to go through the same steps to reproduce your results because the database has been updated?

L. 133: Are these 609 'annualized' cost entries counted by year or by reported cost (e.g. does your example in L. 125 count as 1 or as 10 entries?). Might be worth to add 'spanning X years of management at Y distinct locations' or something like that?

L. 138-141: See main comment (1) - I checked the Leroy 2022 reference and the R package help, but still cannot see how any of this overcomes the fundamental problem that if you have a cost reported as $20 million between 1991 and 2000 (your L. 125 example), the actual cost per year will simply be assumed to be $2 million for every year (if I understand the function 'expandYearlyCosts' correctly?). That is a reasonable assumption to make to average annual costs for comparisons - but not if you want to model the change of annual costs over time (years). There may be sufficient data that are reported on an annual basis for such trends to be estimated, but I would not recommend that with data that have been averaged over many years. From Fig. 2 it also seems that such a model should not start before 1975 (or later) given the paucity of data before then?

L. 186-192: see main comments above: I think this could be very simply stated as 'reported costs increased from X (1930-1950) to Y (2010-2020)' without the need for any models or projections which do not seem to address/overcome the main limitations of the data (e.g. reporting rates will have fundamentally changed between 1930 and 2020, and it is unclear how data averaged over many years can inform temporal increases).

L. 219: is this the 'annualized' cost or the total across all years? Adding eradication operations such as South Georgia, Campbell, Henderson, Palmyra, all Mexican islands etc. must have cost >$23 million? The database on island invasive species eradications lists 1128 rodent eradications (http://diise.islandconservation.org/, much higher than your n=89), and while I realize that most of those may not have costs reported, the number feels far too low? Might be an opportunity to extrapolate actual vs. reported costs?

L. 183-184 and L. 244-245: Maybe reword that these are just the REPORTED costs. Globally speaking, the cost of $38 million / year is actually minute compared to some other expenditures that our governments make. In the Discussion, it would be good to put these costs into perspective (see for example McCarthy et al. (2012) Financial costs of meeting global biodiversity conservation targets: Current spending and unmet needs, Science, 338(6109), 946– 949. Costanza et al. (1997) The value of the world's ecosystem services and natural capital. Nature, 387, 253– 260.).

L. 249: I don't disagree that the costs may have increased, but I don't think that your data can 'undeniably' demonstrate an 'exponential' increase.

Figs. 3-5: Please clarify whether these costs are 'annual' or the total accumulated reported costs. If the latter, then how can costs be compared that might be accumulated over 10 years for one species/sector/geography and only 3 years in another?

If you have any questions about this review, please feel free to contact me any time:
[email protected]

(and disregard the erroneous spelling of my name, which seems to be a PeerJ system error!)

Reviewer 2 ·

Basic reporting

Review of Economic Costs of Invasive Rodents Worldwide: The Tip of the Iceberg
Introduction
This paper aims to estimate the global economic costs of invasive rodents. For that purpose, the authors combine economist cost data from the Invacost Database and some specific complementary sources from the existing literature. The main findings are:
• Reported costs of rodent invasions were significantly increasing over time, they reached a total of 30 US$ 3.6 billion between 1930 and 2022 (annually US$ 38.7 million).
• The highest cost reported was for muskrat Ondatra zibethicus (US$ 377.5 million), then 32 unspecified Rattus spp. (US$ 327.8 million), Rattus norvegicus specifically (US$ 156.6 million) and 33 Castor canadensis (US$ 150.4 million).
• Of the total costs, 87% were damage-related, mainly affecting agriculture and reported in Asia (60%), Europe (19%) and North America (9%).
• The paper highlighted obvious cost underreporting with only 99 documents gathered globally, 36 clear taxonomic gaps, and skewed breakdowns of costs among regions, industries and contexts.
• These findings shows that available information represents a big underestimation of the global costs incurred.

Comments
The empirical section looks solid and clear and the paper ready for publication after some minor changes. In particular, the authors should provide a footnote that they are willing to share their data set in Excel with those who wish to replicate the results of the study.

Experimental design

It's Ok.

Validity of the findings

It's OK.

Additional comments

no comments

---

## Round 0.2 · Minor Revisions

Reviewer 1 has requested some minor changes to improve the current version of the submitted paper. Please incorporate and re-submit. Thanks.

·

Basic reporting

This manuscript synthesizes the reported costs associated with invasive rodents worldwide, and has improved considerably from the previous submitted version.

Overall, the manuscript is publishable and I only have a few minor suggestions for edits:

L. 36-38: Although it is likely evident to the authors, and extensively acknowledged in the Discussion, the fact that the REPORTED COSTS are probably only a small fraction of true costs could be worded more powerfully than in this sentence. When people in the media pick up this paper they will take the numbers as literal values and compare them with other 'costs' in society (87 million is very little!), but if you include a headline figure of 'might actually exceed costs of COVID-19', or something in the billions, it is more likely that the media will take notice.

L. 143: delete 'using'

L. 212: do you mean 'highest'? If 'higher', then higher than what?

L. 253-254: I appreciate that you have revised this sentence based on my previous comments, but unfortunately I still do not agree with this statement. The reported costs have increased. Through some clever accounting you may have factored out inflation (e.g. all in 2017 USD), and you even somehow account for 'research effort' (I admit this is beyond my expertise), but I still think that the conclusion that this indicates "ongoing increase in the rates of biological invasion" cannot really be supported by any of the data presented. Everything has increased since 1930, including trade volume and connections, which makes it likely that invasion rates may have increased. But reported costs are not a reliable and straightforward proxy for invasion rates. There is nothing wrong with stating that costs have increased - whether that is due to increased invasion rates or increased damages from the already present invaders isn't something you can disentangle, but I don't think you need to?

L. 266: 'current research' should be 2 words

L. 313: delete 'known'

L. 349: DALY only defined in L. 352

Thank you very much for acknowledging my review in the paper - that is much appreciated!

Best regards,

Steffen

Experimental design

NA

Validity of the findings

valid

Reviewer 2 ·

Basic reporting

I went through the new version of the article.
Everything is fine.
Regards

Experimental design

OK.

Validity of the findings

Ok

Additional comments

None

---

## Round 0.3 · accepted · Accept

The minor comments by the first reviewer have been incorporated and the second reviewer has already accepted the paper. In my opinion, the paper is OK in its current version and can be accepted for publication.